# Effects of Pulsed Electric Field on the Physicochemical and Structural Properties of Micellar Casein

**DOI:** 10.3390/polym15153311

**Published:** 2023-08-04

**Authors:** Ahmed Taha, Federico Casanova, Martynas Talaikis, Voitech Stankevič, Nerija Žurauskienė, Povilas Šimonis, Vidas Pakštas, Marijus Jurkūnas, Mohamed A. E. Gomaa, Arūnas Stirkė

**Affiliations:** 1State Research Institute Center for Physical Sciences and Technology, Saulėtekio al. 3, LT-10257 Vilnius, Lithuaniaarunas.stirke@ftmc.lt (A.S.); 2Food Production Engineering, National Food Institute, Technical University of Denmark, 2800 Lyngby, Denmark; feca@food.dtu.dk; 3Department of Food Science, Faculty of Agriculture (Saba Basha), Alexandria University, Alexandria 21531, Egypt; 4Micro and Nanodevices Laboratory, Institute of Solid State Physics, University of Latvia, Kengaraga Str. 8, LV-1063 Riga, Latvia

**Keywords:** casein micelles, pulsed electric field (PEF), FTIR, secondary structure, Raman spectroscopy

## Abstract

Pulsed electric field (PEF) as a green processing technology is drawing greater attention due to its eco-friendliness and potential to promote sustainable development goals. In this study, the effects of different electric field strengths (EFS, 0–30 kV/cm) on the structure and physicochemical features of casein micelles (CSMs) were investigated. It was found that the particle sizes of CSMs increased at low EFS (10 kV/cm) but decreased at high EFS (30 kV/cm). The absolute ζ-potential at 30 kV/cm increased from −26.6 (native CSMs) to −29.5 mV. Moreover, it was noticed that PEF treatment leads to changes in the surface hydrophobicity; it slightly increased at low EFS (10 kV/cm) but decreased at EFS > 10 kV/cm. PEF enhanced the protein solubility from 84.9 (native CSMs) to 87.1% (at 10 kV/cm). PEF at low EFS (10 kV/cm) intensified the emission fluorescence spectrum of CSMs, while higher EFS reduced the fluorescence intensity compared to native CSMs. Moreover, the analysis of the Amide Ι region showed that PEF-treated CSMs reduced the α-helix and increased the β-sheet content. Raman spectra confirmed that PEF treatment > 10 kV/cm buried tyrosine (Tyr) residues in a hydrophobic environment. It was also found that PEF treatment mainly induced changes in the disulfide linkages. In conclusion, PEF technology can be employed as an eco-friendly technology to change the structure and physiochemical properties of CSMs; this could improve their techno-functional properties.

## 1. Introduction

Food proteins serve crucial functions in food manufacturing, human nutrition, and the nutraceutical sector. Dairy proteins possess exceptional functional characteristics and exhibit elevated nutritional values [1,2]. Bovine milk proteins consist of whey proteins (20%) and caseins (80%), and they are widely utilized as raw materials in various food products [3]. Dairy proteins are essential in numerous food products as a primary ingredient, necessitating exceptional functional characteristics such as superior solubility, enhanced emulsifying capabilities, and improved foaming and gelling properties [4]. To enhance the functional attributes of dairy proteins, it is necessary to implement various modification techniques to their structural and conformational characteristics.

Several techniques were applied to alter the structural and functional properties of proteins, including ultrasound [5], high-pressure processing [6], and microwave [7]. Additionally, pulsed electric field (PEF) has become increasingly utilized in various food applications as an emerging green technology [8,9]. It has mainly been used to inactivate enzymes and microorganisms [10]. In addition, a number of studies have found that PEF can alter the structure and technological aspects of proteins [11,12,13,14]. The influences of PEF treatment (19.2–211 µs, 30–35 kV/cm) and different heat treatments (30–75 °C) on the functional and structural characteristics of whey protein isolate (WPI) were studied by Sui et al. [15]. PEF treatment did not significantly change the free-SH group content, surface hydrophobicity, and protein unfolding. It was found that similar droplet sizes (~4 μm) were obtained for PEF-induced WPI (30 kV/cm) and heat-induced WPI (72 °C for 15 s) emulsions [15]. However, low EFS (22.7–24.2 °C, 500 V/m) altered the tertiary structure of BSA. Bekard and Dunstan found that the structural changes possibly occurred because of the perturbation of hydrogen bonds in BSA’s native structure [16]. Rodrigues et al. investigated the impacts of a moderate electric field (MEF) on the structures of whey proteins and β-lactoglobulin [17,18]. For β-lactoglobulin, the authors concluded that MEF (0–10 V/cm) improved surface hydrophobicity and changed Trp fluorescence spectra and the secondary structure. The combination of MEF and heat treatment (up to 70 °C) boosted the structural changes of β-lactoglobulin. For WPI, MEF combined with ohmic heating decreased the viscosity and formed smaller aggregates with less thiol groups content.

Caseins (80% of total milk protein) comprised several subunits, including αs1 (45% of total casein), αs2 (12%), β (33%), and kappa (k, 10%) caseins [19]. They are widely utilized as a raw ingredient in several food products. Ultrasound and high-pressure treatments were utilized to change the technological and structural characteristics of caseins. For example, ultrasonication (20 kHz, 0.75 W/mL) reduced the particle size and increased the solubility (>95%) of casein powders (MCP) [20]. In another study, the authors concluded that sonication (30% amplitude, 20 kHz) decreased the solubility and particle size and improved thermal stability of casein micelles at pH 4–8 [21]. Protein tertiary structures were altered by ultrasound, which exposed the hydrophobic residues to the surface of the protein molecules. As a result, protein flexibility increased and the interfacial tension at the oil/water interface decreased, which enhanced proteins’ emulsifying ability [22]. During the treatment of CSMs at high pressure, the structure depends on the magnitude of pressure. If the pressure is low (100–200 MPa), the structure does not change significantly. However, at higher pressures (>400 MPa), the hydrophobic interactions are disrupted, resulting in a reduction in the average size of CSMs [23]. While the effects of other processing techniques, such as ultrasound and high-pressure processing, on the structure of caseins were extensively investigated, there is a lack of knowledge on the PEF effects on CSMs.

Based on the provided information, we inferred that PEF treatment could alter the structure of CSMs. However, the impacts of PEF on the structural characteristics of CSMs are yet to be studied. Therefore, this study aimed to examine the structural modifications of CSMs following PEF treatment (0–30 kV/cm). The results of this study could help in better understanding the effects of PEF on the structural properties of milk proteins. 

## 2. Materials and Methods

### 2.1. Materials

Casein micelles (CSMs, 92% protein) were purchased from VWR Chemicals (Leuven, Belgium). Fluorescein isothiocyanate was obtained from Fluorochem Ltd. (Hadfield, UK). β-mercaptoethanol, sodium hydroxide, and sodium chloride were acquired from AppliChem GmbH (Darmstadt, Germany). Sodium dodecyl sulfate (SDS) and Coomassie blue dye (G250) were acquired from Sigma Aldrich (St. Louis, MO, USA). 8-Anilino-1-naphthalenesulfonic acid (ANS) was purchased from Cayman Chemical Company (Ann Arbor, Michigan, USA). Hydrochloric acid, phosphoric acid, methanol, and ethanol absolute were obtained from Merck (Darmstadt, Germany).

### 2.2. PEF Treatment

In potassium phosphate buffer (0.1 M, pH = 7), a protein suspension (1%, *w*/*v*) was prepared and stirred overnight. The protein suspension (0.5 mL) was transferred to the Fisher electroporation cuvette (2 mm gap, Thermo Fisher Scientific Inc., Waltham, MA, USA). The conductivity of the samples was 6.5 mS/cm. PEF treatment was conducted under the following conditions: The applied voltage was 2, 4, and 6 kV (10, 20, and 30 kV/cm, respectively) for 10 pulses; the current values were 360, 680, and 1060 A (for 10, 20, and 30 kV/cm, respectively); the pulse duration was 480 ns (evaluated from the time constant of the exponent); and the pulse shape was an exponential decay pulse (Figure 1).

The PEF was generated using a generator (designed and built at the Center for Physical Sciences and Technology, Vilnius, Lithuania) consisting of a high-voltage source (maximum voltage of 20 kV) and capacitor bank (100 nF, max voltage of 30 kV). The maximum reachable electric field using a 2 mm cuvette is 100 kV/cm. The cuvette is placed in a holder containing two flexible metal contacts. The current measurement was performed using a shunt resistor connected in series with the cuvette, and the voltage was measured using a high-voltage divider. The generated pulse has an exponential shape with time constant τ=RC, where R is the equivalent circuit resistance and C is the capacitance of the used capacitor. As a result, the effective PEF treatment duration is dependent on the solution’s resistance and is easily controllable. In Figure 2, the electric circuit of the electric field-generating system is displayed.

### 2.3. Particle Size and ζ-Potential

A Malvern Zetasizer analyzer (Nano ZS, Malvern Instrument Co., Ltd., Worcestershire, UK) was utilized to estimate the ζ-potential and particle size of control and PEF-induced CSM samples. Refractive indices for the diluted samples (1:20) were determined at 1.45 for the protein and 1.33 for the dispersant [24].

### 2.4. Turbidity

The CSMs solution was diluted (1:10), and the adsorption values were measured at 600 nm using a UV–visible spectrophotometer (Halo RB-10, Dynamica Scientific Ltd., Kirkton Campus, Livingston, UK) [25].

### 2.5. Protein Solubility

To determine the solubility of CSMs, each sample (1 mL) with a 10 mg/mL initial protein content (P_C_) was centrifuged at 12,000× *g* for 30 min. Then, to determine the protein concentration in the supernatant (P_S_), the Bradford protein assay was utilized. The protein solubility (%) was estimated following Equation (1) [26].
(1)$$\mathrm{Protein}\;\mathrm{solubility}(\%)=\frac{P_{S}}{P_{C}}\times 100\%$$

### 2.6. Surface Hydrophobicity (Ho)

The H_o_ values of the control and PEF-treated CSMs were examined using ANS (1-anilino-8-naphthale-nesulfonate) as a fluorescence probe. Different protein concentrations (0.1, 0.2, 0.5, and 1 mg/mL) were prepared. 80 μL ANS solution (8.0 mM) was mixed with 3 mL of each protein concentration. The combinations were then placed in a dark place for thirty minutes. The fluorescence intensities were then measured using a PerkinElmer LS 50B spectrometer (PerkinElmer, Waltham, MA, USA) with excitation and emission wavelengths of 390 nm and 470 nm, respectively, and a slit width of 5 nm. The slope of the protein concentration and fluorescence intensity was used as the surface hydrophobicity [27].

### 2.7. Intrinsic Fluorescence Spectroscopy

0.1 M phosphate buffer (pH 7.0) was used to modify the protein concentration to 0.5 mg/mL. A 10 mm (3 mm path-length) quartz cuvette received 3 mL of each sample. An LS 50B spectrometer from PerkinElmer was used to record the intrinsic fluorescence spectra. 10 nm was the length of the emission and excitation slits. At 290 nm, excitation wavelengths and fluorescence emission spectra between 300 and 450 nm were captured [28].

### 2.8. Scanning Electron Microscopy (SEM)

A Hitachi SU-70 SEM device (Minato-ku, Tokyo, Japan) was used to capture the SEM micrographs to study the morphology of control and PEF-treated CSMs. Samples of CSM powder were gold-sputtered to produce a thin covering. The study was performed at an accelerated voltage of 2 kV and magnifications of 250 and 2000×.

### 2.9. Fourier Transform Infrared Spectroscopy (FTIR)

An infrared spectrometer (Spectrum 100, PerkinElmer, Norwalk, CT, USA) was used to scan two milligrams of freeze-dried CSM samples. With a resolution of 4 cm^−1^ and 32 scans, the IR spectra of PEF-treated and untreated CSM were obtained from 400 to 4000 cm^−1^. Peakfit software (Peakfit 4.12, Seasolve Software Inc., Palo Alto, CA, USA) was used for identifying the hidden peaks in the amide I (1600–1700 cm^−1^) region in order to gain knowledge on the secondary structural changes of proteins.

### 2.10. Raman Spectroscopy

Raman spectra were collected using an Echelle-type spectrometer, RamanFlex 400 (PerkinElmer, Shelton, CT, USA), with a 785 nm laser excitation source, a thermoelectrically cooled CCD detector (–50 °C), and fiber-optic cable. The laser beam (100 mW) was focused on a 200 µm diameter spot on the CSM powder samples dispersed on a Tienta steel substrate (SpectRIM, Merck, Rahway, NJ, USA) at room temperature with an acquisition time of 1800 s. The ratio of double peaks near 850 cm^−1^ and 830 cm^−1^ (I_850_/I_830_) was calculated to study the hydrogen bonding microenvironment on phenolic hydroxyl groups [26].

### 2.11. Statistical Analysis

The experiments were completed in triplicate. The data are presented as means and standard deviations. Statistical analysis of the raw data was completed via SPSS (IBM SPSS Statistics, version 25, SPSS Inc., Chicago, IL, USA). The means’ statistical significance was investigated using a one-way ANOVA test and Duncan’s test (*p* < 0.05).

## 3. Results and Discussion

After PEF treatment, we observed a slight and gradual rise in temperature (Table 1). Specifically, the temperature increased from 20.3 °C (before PEF treatment) to 24.2 °C after applying PEF at an EFS of 30 kV/cm. This increase, though noticeable, can be considered minimal, mainly because casein exhibits a higher resistance to heat treatment compared to other milk proteins. It is worth noting that casein typically requires heat treatment at temperatures exceeding 100 °C to induce significant structural changes. Therefore, the observed temperature rise resulting from the PEF treatment can be seen as a modest effect and emphasizes casein’s stability under these conditions [29,30].

### 3.1. PEF Effects on the Particle Size and ζ-Potential

The particle size is a critical parameter affecting the solubility and other techno-functional characteristics of protein molecules. The influences of PEF on the ζ-potential and particle size of CSMs are presented in Table 1. The particle size of protein molecules considerably increased from 266.8 to 276.2 nm (at 10 kV/cm). However, increasing the EFS to 20–30 kV/cm decreased the particle size of proteins. The size distribution of control and PEF-treated CSMs exhibited a bimodal distribution; this was similar to the reported size distribution of β-casein [31]. The peak values ranged from 30–100 nm (peak 1) and 200–1000 nm (peak 2). The peaks of CSMs treated at 10 kV/cm (Figure 3) shifted towards larger particle sizes, confirming the formation of larger aggregates. It was recently confirmed that a MEF (8–10 V/cm) formed larger aggregates of soy protein isolate [32]. This could occur due to the partial unfolding or aggregation of protein molecules [27,33]. Interestingly, increasing the EFS (higher than 10 kV/cm) resulted in smaller particle sizes compared to native protein and CSMs treated at 10 kV/cm. PEF treatment could reduce protein size by disrupting hydrogen bonding and electrostatic and hydrophobic interactions [34]. However, the increase in particle size at 10 kV/cm and the decrease at higher EFS need more investigations to verify and understand the potential causes. ζ-potential indicates the electric charge characteristics of proteins. The findings revealed that the absolute ζ-potential value increased from 26.6 (native CSMs) to 30.1 mV following treatment at 10 kV/cm. Similarly, the absolute ζ-potential value of soy protein isolates increased after MEF treatment (4 V/cm) [32]. This could be due to the unfolding of CSMs, which could lead to the exposure of more hidden polar groups to the surface, increasing the surface’s negative charges.

### 3.2. Protein Solubility and Turbidity

Protein solubility is an essential factor affecting functional characteristics, including gelling, emulsifying, and foaming abilities [35]. Several factors can influence protein solubility, including amino acid profile, molecular weight, hydrogen bond content, as well as the hydrophobic and hydrophilic groups’ contents on the surface [27,36]. The solubility improved from 84.9 to 87.1, 86.4, and 86.6% following PEF at 10, 20, and 30 kV/cm, respectively (Table 1). PEF facilitated the formation of hydrophobic interactions among unfolded proteins, forming soluble protein aggregates [37]. After PEF treatment, the increased protein solubility was probably due to the molecular polarization and improved dielectric constant, which resulted after PEF treatment at high EFS. Similarly, PEF treatment (18 kV/cm) of myofibrillar protein was suggested to produce free radicals that might disrupt non-covalent bonds and electrostatic interactions, increasing protein solubility [38]. The size, quantity, and aggregation of suspended proteins are indicated by turbidity [39]. The turbidity of CSMs treated at 10 kV/cm increased significantly compared to native samples (Table 1). The aggregation and the higher particle size may enhance the amount of light absorption and scattering, improving the turbidity [40]. These findings show that PEF (10 kV/cm) could facilitate the formation of hydrophobic interactions and new intermolecular disulfide (S-S) bonds [12,37]. The protein suspensions’ turbidity probably increased due to the growth of aggregates and the rise in particle size, which also increased the amount of diffuse light reflection [32]. These findings agreed with the particle size findings (Table 1).

### 3.3. Tertiary Structure and Surface Hydrophobicity (Ho)

The intrinsic fluorescence spectra indicate the microenvironment’s polarity changes in aromatic amino acids [41]. Thus, the spectra of CSMs before and after PEF treatment were obtained to examine the alterations in the tertiary structure [42]. The fluorescence intensity increased (from 829.5 to 855.6) after treating CSMs at 10 kV/cm but slightly decreased after increasing the EFS higher than 10 kV/cm (Figure 4). When tryptophan residues are positioned in a hydrophobic or non-polar environment, fluorescence intensity increases. However, when tryptophan residues are exposed to a hydrophilic, polar environment, fluorescence is quenched [43]. The increase in peak intensity could occur due to the transformation of active residues into a hydrophobic environment inside the newly developed aggregates [44]. PEF treatment at 10 kV/cm could cause alterations in the conformational structure of proteins, exposing more aromatic amino acids to the protein molecule surface. Moreover, the *λ_max_* slightly blue-shifted from 345.5 nm (native protein) to 344.5 nm (10 kV/cm), indicating that the hydrophobicity around the fluorophore increased [45]. PEF treatment exposed more hidden hydrophobic areas due to partial unfolding [46]. At a pH of around 7, many hydrophobic amino acids have negative charges on their surfaces. Thus, the absolute ζ-potential value increased after PEF treatment, probably because of the exposure of more hydrophobic regions [47]. Exposed hydrophobic regions with more negative amino acids could increase the absolute ζ-potential value [48]. These findings were in line with the ζ-potential (Table 1) and Raman spectroscopy results.

### 3.4. Microstructure

Figure 5 shows the surface morphology of native casein and PEF-treated casein. It could be seen that compact structures with large aggregates were presented in native and 10 kV/cm treaded CSMs. Additionally, the surface appears to be intact, without any visible cracks or fissures. In contrast, in the SEM images of the casein samples treated with PEF, disrupted surface structures with visible cracks and fissures were visible. As the voltage of the PEF treatment increased from 10 to 30 kV/cm, the severity of these disruptions became more pronounced. At higher EFS, the SEM image of the PEF-treated casein sample shows more severe disruptions to the surface structure of the protein, with more visible smaller aggregates with more sheet-like structures compared to native and 10 kV/cm tread CSMs. These SEM micrographs suggest that PEF treatment has caused significant changes in the surface structure and morphology of casein. The degree of disruption to the protein surface increased as the voltage of the PEF treatment increased, indicating that higher voltage treatments cause more severe structural changes.

### 3.5. Secondary Structure

FTIR is a widely used method for determining proteins’ secondary structures. FTIR data can be correlated with the vibrational changes of proteins’ chemical interactions [49]. The FTIR spectrum ranging from 110 and 1700 cm^−1^ typically provides vital information on the structure of polypeptides, with the amide I band (1600 and 1700 cm^−1^) being the most informative. The C=O stretching vibrations of the protein backbone are reflected in the amide I band [50]. Amide I is evaluated as the “fingerprint” of secondary protein structures. It is sensitive to alterations in secondary protein structures and hydrogen bonding [51,52]. The IR Band frequencies of 1605–1611, 1618–1630, 1685–1690, 1630–1645, 1652–1666, and 1670–1675 are assigned to the side chain, intramolecular, aggregated β-sheet, random coil, α-helix, and β-turn, respectively [53,54].

The IR spectra of control and PEF-induced proteins are illustrated in Figure 6, and the calculated contents of secondary structures are presented in Table 2. The peaks in region 1050–1100 cm^−1^ can be credited to the colloidal calcium phosphate’s phosphate stretching. The gradual disappearance of some peaks in this region after PEF treatment could be because of the unfolding of CSMs, dissociation of colloidal calcium phosphate, and shielding of calcium ions [55]. The secondary structure percentages of CSMs were calculated using the Peakfit software for peak deconvolution. Following the second derivative method, the hidden peaks in the amide I area were identified automatically. The peak positions were fitted using the Gaussian model to estimate the presence of secondary structures. The area of each fitted peak was divided by the entire area of the amide I peak, multiplied by 100, to get the percentage of each secondary structure [56].

Compared to native CSMs, the PEF-treated (10–30 kV/cm) CSMs exhibited a significant decrease in the α-helix (from 33.5 to ~20.5%) content as an extra peak appeared in the α-helix region (at 1666 cm^−1^) of native CSM FTIR-fitted peaks (data not shown). The β-turn content increased from 16.6% (native CSMs) to 18.5–20.9% (PEF-treated CSMs). Also, the β-sheets contents increased from 30.5% before PEF treatment to 39.7, 37.5, and 37.6% (for CSMs treated at 10, 20, and 30 kV/cm, respectively). The random coil content decreased from 12.7 before treatment to 10.8% after PEF treatment at 10 kV/cm but increased to 15.8% at 30 kV/cm. Zhang et al. concluded that sonication reduced the α-helix content of micellular casein while increasing the random coil and β-sheet contents. Due to an increase in the concentration of various types of β-sheets, more hydrophobic groups may be exposed to the protein molecules’ surface [57]. The examination of surface hydrophobicity verified these findings, where the surface hydrophobicity and β-sheet content of CSMs treated at 10 kV/cm were higher than other samples. Similar findings were also reported with ultrasound pre-treated casein. The authors found that sonication altered the secondary structure as the random coil and β-turn contents were boosted and the α-helix content was reduced [51]. Electric energy could be absorbed by some polar residues of proteins, producing free radicals. The produced free radicals could disrupt several protein-protein interactions (i.e., Van der Waals, hydrogen and disulfide bonds, and hydrophobic and electrostatic interactions). This may result in partial unfolding of the α-helix region followed by random coil and β-turn formation [14,51]. No obvious changes in the β-sheet and α-helix contents were observed among all PEF-treated CSMs. Melchior et al. [58] researched the impacts of MEF (1.65 kV/cm) on the secondary structure of pea proteins. They observed considerable secondary structure changes at a lower number of pulses, while the increase in pulse numbers did not extensively change the pea protein’s secondary structures. This reveals that the bound charges and energy landscape alterations induced by PEF do not gradually increase after increasing the EFS, at least in the range used in our study [58]. PEF treatment at lower EFS could induce both chemical (aggregation and secondary structure) and physical (increase in particle size) changes. At higher EFS, strong electric shocks might produce smaller soluble particles (Table 1) but do not significantly change the secondary structures of proteins compared to proteins treated at lower EFS.

### 3.6. Raman Spectroscopy Analysis

Figure 7 presents the Raman spectra of CSMs prior to and following PEF treatment. The variations in the frequency and scattering intensity of the Raman spectrum can be used to infer the conformational changes of CSMs [59]. The intensity of the Raman band at 430–550 cm^−1^ sharply increases after PEF treatment; this band is assigned to the disulfide ν(S–S) bond [26,60]. The cysteine disulfide-related spectral modes appear in the 450–700 cm^−1^, range with the exact frequency strongly dependent on the conformation of the CCSSCC moiety [61]. This might be related to the S–S bond isomerization reaction; however, some contribution from Amide VI may also be present. The mode near 710 cm^−1^ is ascribed to stretching vibration of the C–S bond in trans conformation, ν_T_(C–S). The mode increased in intensity in 10 kV/cm samples compared to native CSMs [62]. Dips near 1074 and 1448 cm^−1^ could be linked with changes in C–C and C–N stretching and CH_2_ deformation vibrational intensity, respectively, whereas the negative-facing mode at 1722 cm^−1^ is due to carbonyl vibration, ν(C=O). A dip at 856 cm^−1^ is related to one of the tyrosine’s doublet modes, which is analyzed in more detail in Figure 8.

Tyrosine (Tyr), whose ring breathing mode Y1 appears as a Fermi doublet around 850 and 830 cm^−1^, indicates a hydrogen bonding state and probes a hydrophobic/hydrophilic environment near the Tyr side chain. The intensity ratio (I_850_/I_830_) value of between 0.7 and 1.0 means that Tyr is exposed to a hydrophobic environment. While the ratio (I_850_/I_830_) is between 0.90 and 1.45, indicating that Tyr existed in polar environments [26]. As shown in Figure 8, increasing the EFS decreased the value of the (I_850_/I_830_) ratio. These findings indicate that PEF treatment buried more Tyr residues in a hydrophobic environment, mainly due to intermolecular interactions [59,63]. However, the changes in intensity ratio were small, most likely because the protein was dried before subjecting it to the Raman spectroscopic analysis. Raman data indicate that the biggest changes induced by PEF occur at the disulfide linkages of the protein, both in the structure of S–S and S–C bonds. Some extent of tyrosine sidechain burying was detected as well.

## 4. Conclusions

The PEF treatment was successfully utilized to modify the physicochemical and structural properties of CSMs. The treatment caused polarization of the protein molecules, leading to a significant increase in the absolute ζ-potential values, from 26.6 mV to 30.1 mV. Additionally, the particle sizes of the CSMs decreased after the PEF treatment, with a reduction from 266.8 nm (for native CSMs) to 257.5 nm at 30 kV/cm. The application of high EFS resulted in a reduction in the particle sizes of CSMs, leading to an increased surface area, which, in turn, enhanced the protein solubility from 84.9% to 86.6%. PEF treatment caused notable changes in the secondary structures of the CSMs, specifically increasing the β-sheet content from 30.5% (for native CSMs) to 39.7% (at 10 kV/cm) of the total secondary structures. Moreover, the PEF treatment resulted in a reduction of surface hydrophobicity, decreasing it from 164.9 (in the case of native CSMs) to 159.4 (at 30 kV/cm). Additionally, the α-helix content decreased from 33.5% to 20.5%. Raman data provide evidence that significant alterations induced by PEF occurred at the disulfide linkages of the protein, affecting both the structure of S–S and S–C bonds. The PEF-induced changes in the physicochemical properties of CSMs occurred mainly due to the polarization effects of PEF treatment, followed by the formation of free radicals, exposure of -SH groups, and formation of hydrophobic and S-S bonds. In this study, we utilized small sample volumes of 0.5 mL, which proved sufficient for conducting a laboratory-scale proof-of-concept demonstration and developing the protocols to be potentially employed in forthcoming research. Our future plans involve creating a Marx-based PEF generator with a semi-automated continuous processing system. In summary, we strongly recommend investigating the impact of PEF treatment on casein’s interfacial, emulsifying, and gelling properties. Moreover, cooperation between research institutes and PEF generator companies is also recommended to design and build fully automated and high-quality industrial-scale PEF generators.

## Figures and Tables

**Figure 1 polymers-15-03311-f001:**
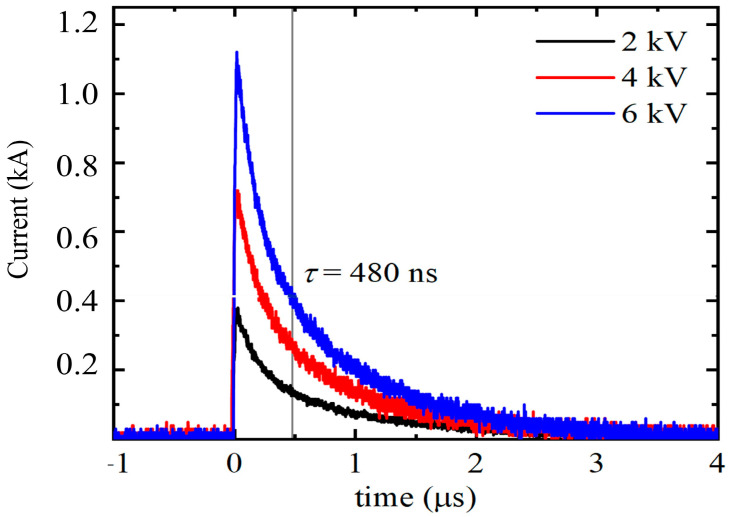
The generated electric pulses of different PEF treatments, which show the applied current and the pulse duration.

**Figure 2 polymers-15-03311-f002:**
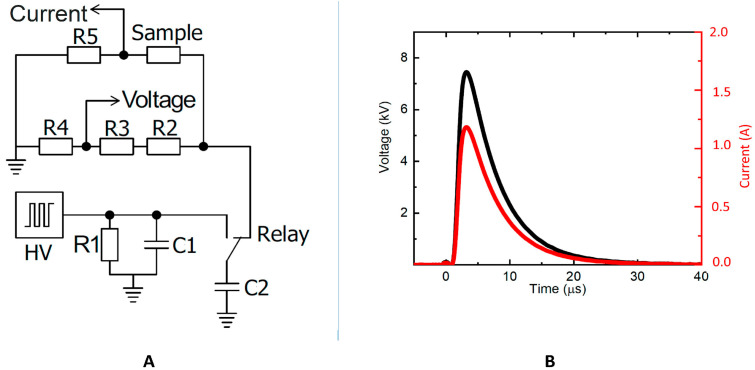
(**A**) Schematic diagram of the homemade electroporator electrical circuit; (**B**) generated electric pulse. R1 = 55 MΩ; C1 = C2 = 100 nF; R2 = R3 = 500 kΩ; R4 = 47 kΩ; and R5 = 0.1 Ω.

**Figure 3 polymers-15-03311-f003:**
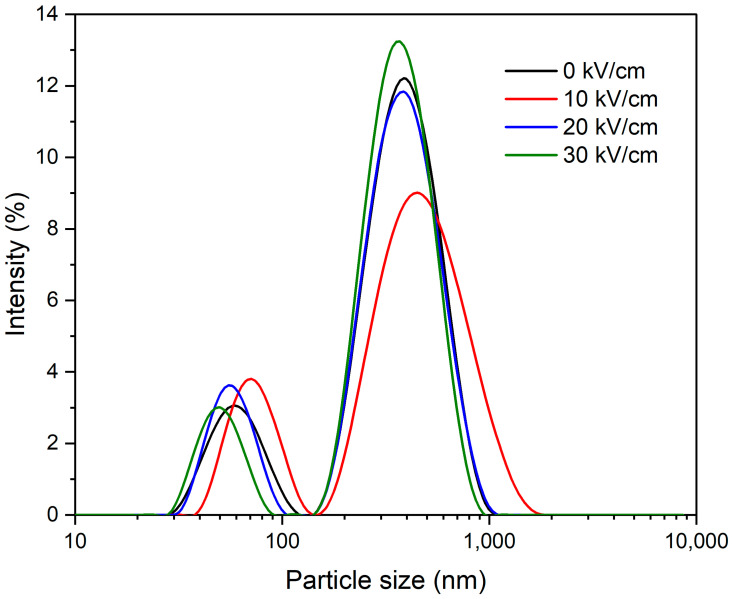
Particle size distribution (PSD) of native (0 kV/cm) and PEF-treated casein micelles at electric field strengths of 10–30 kV/cm.

**Figure 4 polymers-15-03311-f004:**
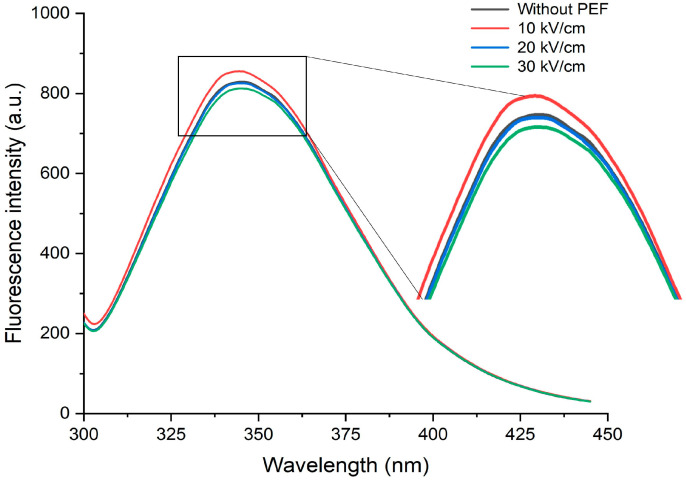
Fluorescence emission spectra of native (0 kV/cm) and PEF-treated casein micelles at electric field strengths of 10–30 kV/cm.

**Figure 5 polymers-15-03311-f005:**
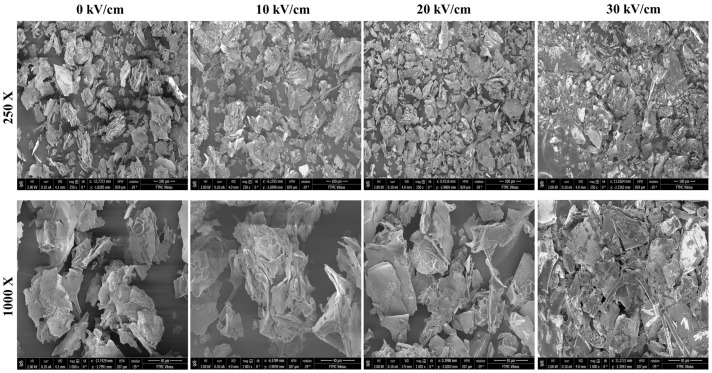
SEM images of native (0 kV/cm) and PEF-treated casein micelles at electric field strengths of 10–30 kV/cm.

**Figure 6 polymers-15-03311-f006:**
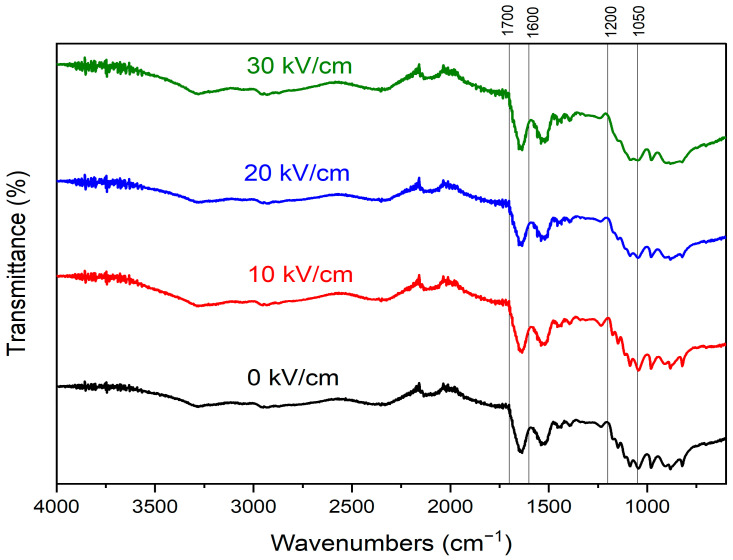
FTIR spectra of native (0 kV/cm) and PEF-treated casein micelles at electric field strengths of 10–30 kV/cm.

**Figure 7 polymers-15-03311-f007:**
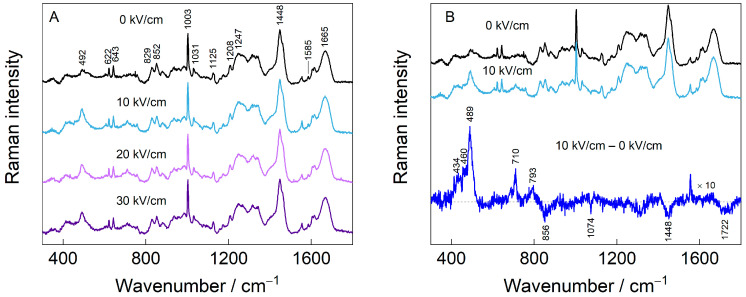
Raman spectra in the fingerprint region (**A**) of native (0 kV/cm) and PEF-treated casein micelles at electric field strengths of 10–30 kV/cm, and (**B**) the difference between native and 10 kV/cm-treated CSMs.

**Figure 8 polymers-15-03311-f008:**
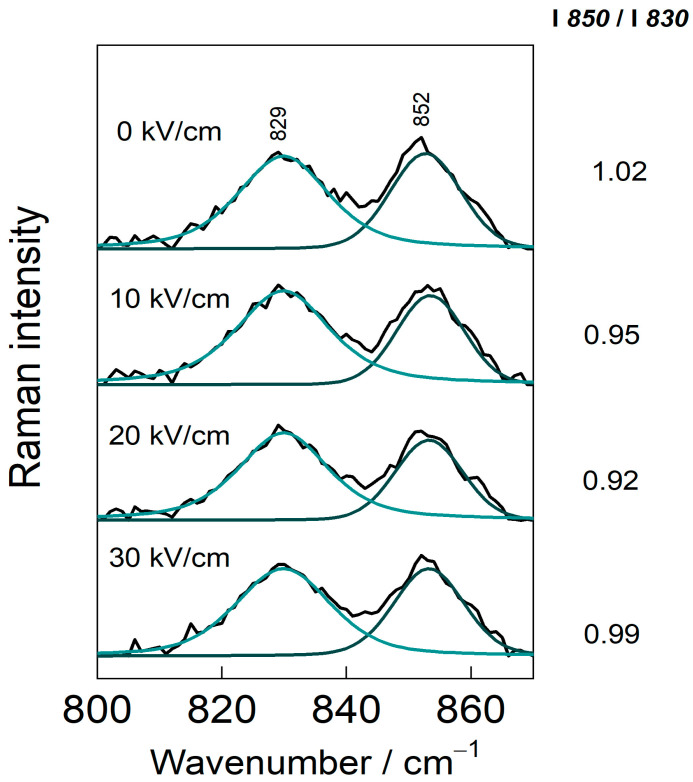
Raman spectra (820–860 cm^−1^) and the calculated ratio of the relative integral intensities at 850 and 830 cm^−1^ of native (0 kV/cm) and PEF-treated casein micelles at an electric field strength of 10–30 kV/cm.

**Table 1 polymers-15-03311-t001:** Particle size (z-average), PDI, ζ-potential, turbidity, protein solubility, surface hydrophobicity (Ho), temperature of native (0 kV/cm), and PEF-treated casein micelles at electric field strengths of 10–30 kV/cm. Means with different letters (^a^, ^b^, ^c^) in each column indicate statistically significant differences among protein samples following Duncan’s analysis (*p* < 0.05).

	Z-Average (nm)	PDI	ζ-Potential (mV)	Turbidity	Protein Solubility	Surface Hydrophobicity (Ho)	Temperature (°C)
0 kV/cm	266.8 ± 2.3 ^b^	0.52 ± 0.03 ^b^	−26.6 ± 0.2 ^b^	0.137 ± 0.001 ^c^	84.9 ± 0.3 ^b^	164.9 ± 1.3 ^a^	20.3 ± 0.3
10 kV/cm	276.2 ± 3.1 ^a^	0.61 ± 0.01 ^a^	−30.1 ± 0.3 ^a^	0.152 ± 0.002 ^a^	87.1 ± 0.2 ^a^	166.0 ± 2.4 ^a^	21.1 ± 0.4
20 kV/cm	258.2 ± 2.9 ^c^	0.56 ± 0.01 ^b^	−29.3 ± 0.8 ^a^	0.141 ± 0.002 ^b^	86.4 ± 0.3 ^a^	160.6 ± 2.7 ^b^	22.3 ± 0.3
30 kV/cm	257.5 ± 3.3 ^c^	0.59 ± 0.02 ^a,b^	−29.5 ± 0.5 ^a^	0.143 ± 0.003 ^b^	86.6 ± 0.4 ^a^	159.4 ± 1.3 ^b^	24.2 ± 0.5

**Table 2 polymers-15-03311-t002:** Total percentage area of secondary structures in the Amide Ι region (1600–1700 cm^−1^) in the FTIR spectra of native (0 kV/cm) and PEF-treated casein micelles at an electric field strength of 10–30 kV/cm. Means with different letters (^a^, ^b^, ^c^) in each column indicate statistically significant differences among protein samples following Duncan’s analysis (*p* < 0.05).

	Peak Area (%)				
	Side Chain	Intramolecular and Aggregated β-Sheet	Random Coil	α-Helix	β-Turn
0 kV/cm	6.5 ± 1.2 ^a^	30.5 ± 2.1 ^c^	12.7 ± 2.3 ^c^	33.5 ± 2.4 ^a^	16.6 ± 2.1 ^c^
10 kV/cm	7.9 ± 0.9 ^a^	39.7 ± 2.5 ^a^	10.8 ± 1.7 ^d^	20.5 ± 1.9 ^b^	20.9 ± 2.6 ^a^
20 kV/cm	7.7 ± 1.3 ^a^	37.5 ± 1.9 ^b^	14.7 ± 1.6 ^b^	20.8 ± 1.5 ^b^	19.1 ± 1.8 ^b^
30 kV/cm	7.4 ± 0.8 ^a^	37.6 ± 2.8 ^b^	15.8 ± 1.9 ^a^	20.5 ± 2.2 ^b^	18.5 ± 2.1 ^b^
Band frequency (cm^−1^)	1605–1611	1618–1630 and 1685–1690	1630–1645	1652–1666	1670–1675

## Data Availability

Data are contained within the article.

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
