# Peer review of "Effects of Pulsed Electric Field on the Physicochemical and Structural Properties of Micellar Casein"

_polymers, 2023, doi:10.3390/polym15153311_

Round 1

Reviewer 1 Report

1.     Physicochemical and Structural Properties are written in the title, but what is the difference between these two properties? Which indicators are Physicochemical Properties? Which are Structural Properties? Please explain.

2.     Abstract, the abstract is cumbersome and logically weak; please be concise. Last sentence: In addition, applying different PEF treatments can modulate the physicochemical properties of CSMs. superfluous sentence.

3.     Line 42-43, a number of studies, but there is only one reference.

4.     Line 49-56, Whose research? Please write the author's name, e.g., Rodrigues et al. studied...

5.     Introduction, the logic and structure of the Introduction section require significant adjustments.

6.     Additional experiments on the properties of casein micelles are needed to confirm the changes in the structure of casein micelles.

7.     All legends in the figure do not need an outer frame.

8.     There is no turbidity determination method in Materials and Methods.

9.     Materials and Methods do not correspond to Results and Discussions. For example, SEM

10.   3.5 Secondary structure, the division range of each secondary structure in the Amide I region (1600-1700 cm-1) should be written.

11.   Table 2, The form frame line is not standardized.

12.   Line 315, 1666 cm -1 should be cm-1.

13.   Conclusions, Reasons for adding important data changes in the Conclusion.

No comments

Author Response

Responses to reviewer #1

General comments:

This work mainly investigated the effects of different pulsed electric field (PEF) strengths (0- 30 kV/cm) generated using a resistance–capacitance (RC) circuit on the structure and physicochemical features of casein micelles (CSMs). This is an original and interesting research article. Researchers have a clear idea of the purpose and characterization of technology. This manuscript has rigorous logic and credible results. Nevertheless, there are still many problems in the abstract, introduction, and conclusion of this manuscript, and there is a lack of research on the characteristics of casein micelles. Not conducive to other researchers references. Hence, a few recommendations for potential improvement in certain parts of the manuscript are given below for the consideration of the authors.

Response:
   We are sincerely grateful for the reviewer's favorable remarks concerning the originality of our research and the credibility of our results. In response to the invaluable and constructive feedback, we have diligently labored on refining the manuscript. We hope this process will lead to a significant enhancement in the overall quality of our work.

Specific comments:

  1. Physicochemical and Structural Properties are written in the title, but what is the difference between these two properties? Which indicators are Physicochemical Properties? Which are Structural Properties? Please explain.

Response:
Thanks reviewer #1 for your valuable comment.

The physicochemical properties can be divided to chemical and physical properties. Physical properties may include the particle size, surface charge, and surface morphology (SEM) while chemical properties may be evaluated using protein solubility, fluorescence spectroscopy, surface hydrophobicity analysis. FTIR and Raman spectroscopy are used to evaluate the structural changes of proteins. Moreover, similar expressions/terms were used in recently published articles (for example, Food Hydrocolloids, Volume 108, November 2020, 106065, https://doi.org/10.1016/j.foodhyd.2020.106065 and Food Chemistry, Volume 402, 15 February 2023, 134265, https://doi.org/10.1016/j.foodchem.2022.134265).

  1. Abstract, the abstract is cumbersome and logically weak; please be concise. Last sentence: In addition, applying different PEF treatments can modulate the physicochemical properties of CSMs. superfluous sentence.

Response:
Thank you, the abstract was improved to make it more concise and superfluous sentence and information were deleted.

  1. Line 42-43, “a number of studies”, but there is only one reference.
    Response:
    Thank you, this reference is a review article in which many studies were mentioned. However, we added more references to support this claim.

  1. Line 49-56, Whose research? Please write the author's name, e.g., Rodrigues et al. studied...
    Response:
    The authors’ names were written in the revised manuscript.

  2. Introduction, the logic and structure of the Introduction section require significant adjustments.
    Response:
    We have made significant changes in the introduction to overcome the logical and structural issues in the revised manuscript.

In the revised manuscript, the 1st paragraph represents general introduction about food proteins, their importance, and the need for structural and functional modifications. The 2nd paragraph represents a general introduction on PEF technology, and PEF applications and effects on different subunits of whey proteins. The 3rd paragraph summarizes the effects of other processing technologies (I.e., ultrasound and high pressure processing) on casein. The last paragraph represents our hypothesis and the aim of the study.

  1. Additional experiments on the properties of casein micelles are needed to confirm the changes in the structure of casein micelles.
    Response:
    Thank you review #1 for your significant comment.

Ι-We applied Fluorescence spectroscopy, FTIR and Raman spectroscopy to study the tertiary and secondary structure of CSMs. In FTIR analysis, we not only describe the IR spectra but applied deconvolution test to further investigate the secondary structure contents. These methods are commonly used for studying the protein structure changes. For example, recently, Xu et al. (2023)* applied FTIR and deconvolution analysis to study the influence of protein concentration on amyloid fibrillation of soy protein. In another study, Sun et al. (2021) ** used FTIR and deconvolution analysis to investigate the secondary structure of whey protein gel.

*https://doi.org/10.1016/j.foodhyd.2023.109085
** https://doi.org/10.1016/j.ultsonch.2021.105810

Ⅱ- We have only 10 days to submit our revised manuscript to the journal so the time is so limited.

Ⅲ- Most of researchers and technicians in our institution laboratories are on their summer vacation and it is not possible to perform extra experiments at the moment.

Thanks to the reviewer for his/her understanding.

  1. All legends in the figure do not need an outer frame.
    Response:
    Thank you, in our original manuscript, outer frames of figures’ legends were removed.

  2. There is no turbidity determination method in Materials and Methods.
    Response:
    We added the experimental details of turbidity measurements in the revised manuscript.

  3. Materials and Methods do not correspond to Results and Discussions. For example, SEM
    Response:
    We have changed the order of SEM measurement in the methods section to solve this issue.

  4. 3.5 Secondary structure, the division range of each secondary structure in the Amide I region (1600-1700 cm-1) should be written.

Response:
We added all secondary structures IR ranges in the discussion part of secondary structure.

  1. Table 2, The form frame line is not standardized.
    Response:
    We have checked and edited the format of the table.

  1. Line 315, 1666 cm -1 should be cm-1.

Response: corrected

  1. Conclusions, Reasons for adding important data changes in the Conclusion.

Response: The conclusion section was rewritten, and significant changes were made in the revised manuscript.

Reviewer 2 Report

This paper presents results from the PEF use to modify the physicochemical and structural properties of casein micelles. There are some issues that should be improved, in order to increase the archival value of the document and interest to readers:

Authors present results from the application of exponential pulse voltages to material inside cuvettes:

1)    What is the justification to choose the PEF protocol range used,

2)    While discussing the results from other authors in the Introduction and during Discussion, authors show references were different pulse shapes are used, e.g. rectangular pulses. Nothing is said about these, and how this can affect the results and the comparisons made, as there is no reference about the pulse shape, which is very important.

3)    The results are based on < 1 mL batch samples tests. What are the effects of using such small quantities, what are the problems of scaling-up.

4)    Values of temperature rise, during pulse, should be presented and what are the effects on the results.

The Conclusion should be rewritten as they are similar to the Abstract, numeric values and limitations should be presented.

No comments

Author Response

Responses to reviewer #2

General comments:

This paper presents results from the PEF use to modify the physicochemical and structural properties of casein micelles. There are some issues that should be improved, in order to increase the archival value of the document and interest to readers:

Authors present results from the application of exponential pulse voltages to material inside cuvettes:

Response:
We would like to thank reviewer #2 for his/her efforts in reviewing our works. The comments from the reviewer should help the authors to improve the quality and readability of our research.

Specific comments:

1)    What is the justification to choose the PEF protocol range used,

Response:
In our preliminary experiments, we conducted tests at an electric field strength of 5 kV/cm, administering 1 to 5 pulses. However, the results indicated no discernible impact on the physicochemical properties of CSMs. As a result, we have made the decision to explore higher electric field strengths of 10, 20, and 30 kV/cm in this research. For consistency, we applied a fixed number of 10 pulses in each electric field strength.

2)    While discussing the results from other authors in the Introduction and during Discussion, authors show references were different pulse shapes are used, e.g. rectangular pulses. Nothing is said about these, and how this can affect the results and the comparisons made, as there is no reference about the pulse shape, which is very important.

Response:
Thank you for this constructive and valuable comment, this is the main limitation of PEF applications. In our recent review article (https://www.mdpi.com/2304-8158/11/11/1556), we highlighted this issue in the conclusion “PEF treatment conditions, such as electric strength, pulse shape, pulse duration, and the type of treatment chamber, have a significant impact on the effects of PEF on the structure and techno-functional properties of proteins”
 and it was mentioned that “The main challenge of PEF applications is that many factors (such as PEF device parameters and external factors, such as conductivity, pH, and concentration of treated solutions) can affect the treatment results. Consequently, studies focusing on thermal, chemical, and biophysical components of PEF effects on protein structures should be conducted until clear mechanisms are elucidated. It is also extremely important that authors provide all the necessary details about treatment conditions so the analog studies can be implemented, and results can be compared between those studies. We recommend referring to the guidelines and recommendations proposed by Cemazar et al. (2018) (Bioelectrochemistry 2018, 122, 69–76) for reporting on PEF applications.”
To conclude, it is hard to compare different PEF applications studies as different PEF parameters including pulse shape, applied voltage, number of pulses, etc. and external conditions such as protein concentration, protein type, solution electrical conductivity, pH, etc., can significantly affect the influences of PEF treatment on protein structure.

In our case, to compare the effect of exponential decay with rectangular pulse shapes, we should fix all other PEF and external parameters to get reliable results.

Thus, we generally report others results and compare it with our results without extensive analysis as different PEF and external conditions were applied.

3)    The results are based on < 1 mL batch samples tests. What are the effects of using such small quantities, what are the problems of scaling-up.
Response:
Thank you, This PEF device is designed at our institution to work with small cuvettes. The small sample volume makes it easier to treat small samples with expensive materials such as phytochemicals. Applying PEF treatment to a small quantity of samples should have similar effect when applying PEF at a large scale or bigger sample volume as the electric field go through the ions in the whole sample from one side of the cuvette to the other side.

In our upcoming research, we will use a Marx pulse generator to generate continuous pulses and the sample will be pumped from a sample container to the cuvette with a pump at specific flow rate and then the treated sample will be collected; this is not a fully automated process. The main problems with the scaling up is the funding; we are a research institute and scaling-up such PEF generators with fully automated process needs a significant amount of funding. Moreover, we realized that using pumps in the PEF processing of protein samples could minimally affect the protein structure especially at high flow rates.

4)    Values of temperature rise, during pulse, should be presented and what are the effects on the results

Response:
We have added the following paragraph to the revised manuscript “After PEF treatment, we observed a slight and gradual rise in temperature (Table 1). Specifically, the temperature increased from 20.3°C (before PEF treatment) to 24.2°C after applying PEF at an EFS of 30 kV/cm. This increase, though noticeable, can be considered minimal, mainly because casein exhibit higher resistance to heat treatment compared to other milk proteins. It is worth noting that casein typically necessitates heat treatment at temperatures exceeding 100°C to induce significant structural changes. Therefore, the observed temperature rise resulting from the PEF treatment can be seen as a modest effect and emphasizes casein's stability under these conditions [25,26]”

  1. The Conclusion should be rewritten as they are similar to the Abstract, numeric values and limitations should be presented.

Response:
We have extensively revised and significant changes were made in the conclusion section based on your valuable comment to improve the quality of our manuscript.
